# An Empirical Model for Predicting the Fresh Food Quality Changes during Storage

**DOI:** 10.3390/foods12112113

**Published:** 2023-05-24

**Authors:** Reham Abdullah Sanad Alsbu, Prasad Yarlagadda, Azharul Karim

**Affiliations:** 1School of Mechanical, Medical and Process Engineering, Queensland University of Technology, Brisbane, QLD 4001, Australia; rehamabdullahsanad.alsbua@hdr.qut.edu.au; 2School of Engineering, University of Southern Queensland, Springfield Central, QLD 4300, Australia; y.prasad@usq.edu.au

**Keywords:** fresh food quality, apple, Australia, firmness, weight loss, experimental investigations

## Abstract

It is widely recognized that the quality of fruits and vegetables can be altered during transportation and storage. Firmness and loss of weight are the crucial attributes used to evaluate the quality of various fruits, as many other quality attributes are related to these two attributes. These properties are influenced by the surrounding environment and preservation conditions. Limited research has been conducted to accurately predict the quality attributes during transport and storage as a function of storage conditions. In this research, extensive experimental investigations have been conducted on the changes in quality attributes of four fresh apple cultivars (Granny Smith, Royal Gala, Pink Lady, and Red Delicious) during transportation and storage. The study evaluated the weight loss and change in firmness of these apples varieties at different cooling temperatures ranging from 2 °C to 8 °C to assess the impact of storing at these temperatures on the quality attributes. The results indicate that the firmness of each cultivar continuously decreased over time, with the R^2^ values ranging from 0.9489–0.8691 for red delicious, 0.9871–0.9129 for royal gala, 0.9972–0.9647 for pink lady, and 0.9964–0.9484 for granny smith. The rate of weight loss followed an increasing trend with time, and the high R^2^ values indicate a strong correlation. The degradation of quality was evident in all four cultivars, with temperature having a significant impact on firmness. The decline in firmness was found to be minimal at 2 °C, but increased as the storage temperature increased. The loss of firmness also varied among the four cultivars. For instance, when stored at 2 °C, the firmness of pink lady decreased from an initial value of 8.69 kg·cm^2^ to 7.89 kg·cm^2^ in 48 h, while the firmness of the same cultivar decreased from 7.86 kg·cm^2^ to 6.81 kg·cm^2^ after the same duration of storage. Based on the experimental results, a multiple regression quality prediction model was developed as a function of temperature and time. The proposed models were validated using a new set of experimental data. The correlation between the predicted and experimental values was found to be excellent. The linear regression equation yielded an R^2^ value of 0.9544, indicating a high degree of accuracy. The model can assist stakeholders in the fruit and fresh produce industry in anticipating quality changes at different storage stages based on the storage conditions.

## 1. Introduction

Fresh food supply chain demands special attention, as plant-based food materials (PBFM) are considered important sources of bioactive compounds and minerals essential for mankind [1,2]. Losses of fruits and vegetables in the food chain are estimated to be about a third of the total global production [3]. During transportation and storage, PBFMs undergo physical and biochemical modifications and a significant outcome of this modification is shrinkage and firmness: a decrease in volume, shape, porosity, and changes in hardness [4]. Of the many adverse effects of shrinkage to be found in the available literature, a particularly significant one is the reduction of rehydration capability, which modifies texture [5].

Due to the consumer emphasis on high quality products, producers and distributors are currently prioritizing the preservation of fresh food quality during the transportation and storage through the utilization of advanced cooling and preservation technologies [6,7,8,9]. As a consequence of this transformation, the supply chains for fruits and vegetables are moving away from the traditional model that involved intermediaries such as wholesalers and progressing toward a more direct and collaborative relationship between farmers and supermarkets. This shift allows supermarkets to establish a reliable supply of fresh produce directly from farms, ensuring consistent quality [10].

This study focuses on the impact of storage and transportation conditions on the quality of fresh produce, using apple as a sample. Although changes in many quality attributes, including biochemical properties, take place during storage, the study investigates only the physical qualities and more specifically the firmness and weight loss of apples under various environmental conditions during the storage and transportation stages. These two parameters are considered crucial and have a strong correlation with other quality indicators [11]. The primary aim of this study is to develop and validate an empirical quality degradation model for predicting changes in quality of fresh produce throughout the supply chain.

The quality of fresh fruits and vegetables can be defined as the extent to which a set of inherent characteristics satisfies customer expectations [12]. This definition is also supported by [13], who describe quality as a dynamic composite of physiochemical properties and consumer perception. Harvest time, picking and handling techniques, storage temperature, and other factors significantly impact the quality. The timing of harvest is particularly critical, as early harvesting leads to changes in flavor, color, size and, storability, while late harvesting results in softening and reduces shelf life of apples [14]. Fruit maturity at harvest time and storage conditions significantly affect fruit firmness, which can be measured using various destructive and non-destructive methods [15]. Despite differences in cultivars, firmness change during storage is a common occurrence [14].

Apple (Malus Domestica) is among the top four most widely produced and consumed fruits worldwide, with an annual global production exceeding 87 million tons [16]. Cultivated in subtropical and tropical environments, the European Apple Inventory lists more than ten thousand apple cultivars, resulting in an extensive range of quality traits [17,18]. Firmness is a key attribute for assessing apple quality, as it is closely associated with water content and is the most noticeable change during shelf life [19]. 

Flesh firmness has been widely used to assess the quality of apples at different temperatures, as demonstrated in experiments by [13] who reported changes in the texture of apples with temperature. Similarly, ref. [20] studied the effect of storage temperature and ripening stage on the firmness of fresh cut tomatoes. Several mathematical models have been utilized to predict the quality indicators and determine the quality status of a product at a given stage. Ref. [21] reviewed the kinetics of thermal softening in foods and found that first-order kinetic expressions were suitable for expressing the degree of softening at a constant temperature. The Arrhenius relationship has been utilized to determine the effect of temperature on the quality of fruits, where the relationship between firmness and temperature is approximately linear [22,23]. The degradation of food quality during storage or transport is determined by storage time (*t*), storage temperature *T*, and other constants such as the gas constant and activation energy. 

Quality is changed over time according to the following Equation (1):(1)$$\frac{dq}{dt}=kq^{n},$$
where *q* represents the quality indicator of the product, *k* is the rate of quality degradation which depends on environment conditions such as temperature, *n* is the order of the reaction which can be either 0 or 1 (zero order and first order reactions). Value of *n* determines whether reaction rate is dependent on the amount of quality (*q*) left. When *n* equals 1 (first-order), the quality of the food decays exponentially because of microbial growth (e.g., fresh meat and fish). Fresh fruits and vegetables follow the zero-order reaction when the quality decays linearly. The quality degradation for both orders is illustrated in Figure 1 [24]. 

Temperature of transport or storage is a critical factor affecting and controlling food quality in the supply chain, and the quality changes can be expressed using the following Arrhenius equation:ln K = ln A − E_a_/RT leading to: K = A × e^[−Ea/RT]^(2)
where *K* is the quality degradation rate, *E_a_* is the activation energy (an empirical parameter characterizing the exponential temperature dependence), *R* is the universal gas constant, *A* is the rate constant, and *T* is the absolute temperature (in Kelvin degrees). Equation (1) can be used to describe the change in firmness of the apple fruit given that firmness depends on the temperature of the fruit of the ambient environment [23].

Based on Equations (1) and (2), quality level of fruit can be predicted at certain time in the supply chain using the initial quality (q_0_), the time t_i_ and the degradation rate k_i_ (depending on the temperature T_i_), leading to:q = q_0_ − ∑k_i_t_i_(3)
for zero-order reaction. Substituting the equations for the rate of degradation we get the following equations to express the degradation of quality in specific time at certain temperature: q(t)= q_0_ − k_0_ t exp(−a/T), t ≤ t_1_(4)
where t_1_ is the delay time before the product being stored or transported. And a is a constant of E_a_/R.

Ref. [25] modelled the quality of tomato based on the change of color under constant and variable temperature. From that model we can observe the following equation:f(q) = f(q_0_) + ρt(5)
where q is the quality attribute, and q_0_ is the initial quality attribute, t is time of storage or transportation, ρ is the reaction rate or slope depending on temperature, and f is a function of q which is here the firmness.

The preservation of fruit quality throughout the food supply chain is crucial for minimizing quality loss and maintaining high nutritional value. Given the importance of fresh produce quality in customer-focused competitive strategies, both suppliers and supermarkets make considerable efforts to maintain the quality of fresh produce to increase profits and reduce wastage. However, there is a lack of studies on experimentally determining the quality changes in the supply chain in Australia. This gap in knowledge presents a significant challenge for effectively selecting appropriate storage technology and transportation conditions. The present study aims to fill this research gap by investigating the impact of storage temperature and duration on apple quality, with firmness being the key quality indicator. Although changes in many quality attributes, including biochemical properties, take place during storage, the study investigates only the physical qualities and more specifically the firmness and weight loss of apples under various environmental conditions during the storage and transportation stages. These two parameters are considered crucial and have a strong correlation with other quality indicators. 

Prediction quality changes in the supply chain is an important tool to minimize the food loss. Traditional models such as the Arrhenius model have limited practical applicability in evaluating fruit quality due to their reliance on difficult-to-determine parameters such as initial quality attributes and degradation rates. The study proposes a multi-regression model that considers four main apple cultivars, which have been developed through extensive experimentation and statistical analysis. This empirical model will accurately predict changes in firmness over time as a function of temperature, allowing managers to estimate quality changes during storage or transportation under different conditions. The developed models provide a more realistic representation of product quality. By gaining a more accurate understanding of the relationship between quality and environmental conditions, this study will help to improve product quality and reduce waste in the fruit supply chain.

## 2. Materials and Methods

An experimental investigation was conducted to examine the changes in quality of locally harvested apples during transportation and storage. The study involved four cultivars of apple namely ‘Golden Delicious’, ‘Granny Smith’, ‘Royal Gala’, and ‘Pink lady’ obtained from a commercial farm in Stanthorpe, located in southern Queensland. Apples were grown using integrated fruit production methods and were harvested at their optimum maturity.

Upon harvesting, the apples were stored in large bins and transported into a cold storage facility, where they were kept until the next harvesting season to meet market demand. Harvesting occurred between early February and late May, with each cultivar being harvested at a different time. The fruits were sorted and only the fault-free apples were placed in the cold storage.

Random sampling was employed to collect 100 apples from each cultivar at the farm. The samples were transported to the laboratories of the Faculty of Engineering (O block) and Faculty of Health (Q block) at the Queensland University of Technology, where they were divided into five groups of 20 samples each and stored for two days at four different temperatures: 2 °C, 4 °C, 6 °C, and 8 °C. The firmness and weight loss were analyzed four times at 12-h intervals during the two-day storage period.

### 2.1. Measuring the Weight Loss

Weight loss was determined by periodical weighing of samples. The weight was measured using a digital balance from A&D company ltd (model FZ-300i, San Jose, CA, USA). Twenty apples from each group were taken for the weight loss test. The weight loss (%) was calculated using the formula:Weight loss (%) = ((Fruit initial weight − Fruit weight after interval)/Fruit initial weight) × 100 

### 2.2. Testing the Firmness

To measure the firmness, a fruit pressure tester (Penetrometer) with a plunger of diameter 8 mm was used. Firmness results were expressed in kg·cm^−2^. Penetrometer is one of the destructive methods used to measure the firmness and maturity of a fruit based on the force placed on a known diameter into a growing medium. The hand operated penetrometer used in this study is made by QA supplies LLC (Norfolk, VA, USA). 

To test the firmness of the apples, several steps were followed. Initially, a random sample of apples was taken from boxes of each size, representing four different cultivars. The apples were then prepared by peeling the skin from opposite sides, ensuring a uniform thickness of the tested area. The peeling process was carried out on the cheek of the apple, between the stem and calyx. Each apple was tested twice, at opposite sides of its largest diameter.

The measurement of firmness was conducted by holding the apple against a bench top and placing the plunger tip of a penetrometer on the fruit’s surface. With a consistent and uniform force, the plunger was pressed into the apple. The firmness reading was recorded, and the same procedure was repeated for the opposite side of the apple. This test was repeated on two additional samples.

Once all the firmness readings were obtained, the next step involved interpreting the results. This included calculating the mean firmness by considering all the readings obtained from the multiple apple samples tested.

Additionally, it was recommended that the temperature of the fruit be maintained at 10 °C or higher during the testing process. This temperature was suggested to ensure practical and accurate results since colder fruits tend to yield higher firmness readings. 

Figure 2 shows the experimental facility and the procedure followed to measure the firmness and weight loss.

### 2.3. Validation of the Model

In order to verify the accuracy of the empirical models formulated, the extrapolated model outputs were evaluated against experimental results. To achieve this, a further set of experiments was conducted using a random selection of four apple cultivars, and initial firmness and weight measurements were obtained. The fruits were subjected to measurement of firmness and weight loss at intervals of 12, 24, 36, and 48 h. To ensure greater precision, the experiment was performed in duplicate. 

### 2.4. Statistical Analysis

In this study, four cultivars of apples were selected, and the experiment was designed using a completely random scheme with a total of 16 samples subjected to four different storage conditions. The data obtained from the experiments were analyzed using SPSS (Statistical Package for the Social Science). An ANOVA test was performed, and a significance level of *p* < 0.05 was achieved for each cultivar. 

## 3. Results and Discussion

In this comprehensive study, one of the objectives was to assess the influence of postharvest storage and transportation temperature on the quality of four distinct fresh apple cultivars. To achieve this, firmness and weight loss measurements were used as the indicators of physical quality. The obtained results are presented in Figure 3 and Figure 4, which depict the variations in firmness and weight loss for each cultivar investigated across a wide range of storage temperatures, specifically 2 °C, 4 °C, 6 °C, and 8 °C.

Figure 3 effectively captures the changes in firmness for each apple cultivar at the four different temperatures. An ANOVA test of the results demonstrated the statistical significance of the data (*p* < 0.05) for each cultivar. The analysis of the data revealed interesting insights into the firmness patterns exhibited by the cultivars under varying storage conditions. Notably, the R^2^ values, which indicate the strength of the relationship between temperature and firmness, ranged from 0.8691 to 0.9489 for red delicious, 0.9129 to 0.9871 for royal gala, 0.9647 to 0.9972 for pink lady, and 0.9484 to 0.9964 for granny smith. These high R^2^ values signify a robust correlation between storage temperature and firmness, underscoring the critical role of temperature control in preserving the quality of apples during postharvest handling. These results are supported by the findings of [26]. All the test data, together with the standard deviation, has been presented in Appendix A. 

Furthermore, Figure 4 presents the weight loss profiles of the apple cultivars over time at different storage temperatures. The data clearly reveal an upward trend in the percentage of weight loss as time progresses. Again, the results were significant (*p* < 0.05) as demonstrated by ANOVA tests. The researchers conducted linear regression analyses to derive equations that quantify the relationship between weight loss and time. The resulting coefficients of determination (R^2^) ranged from 0.8407 to 0.9944 for pink lady, 0.7889 to 0.9971 for red delicious, 0.7531 to 0.9733 for granny smith, and 0.7286 to 0.9942 for royal gala. These coefficients provide a quantitative measure of the influence of temperature on weight loss for each cultivar, highlighting the significance of temperature management in mitigating weight loss and prolonging the shelf life of apples. Ref. [27] also reported similar weight loss during storage of pomegranate. 

Analyzing the findings in greater detail, it is evident that both firmness and weight loss exhibit a clear declining trend over time. Before storage, the initial firmness of random samples was measured as a baseline for comparison. Notably, at a storage temperature of 2 °C, all cultivars displayed the highest firmness compared to the other temperatures investigated. The rate of firmness decline varied among the four cultivars, reflecting the inherent differences in characteristics between them. Pink lady consistently exhibited the highest firmness, measuring 8.69 kg·cm^−2^ at the start of storage and maintaining a firmness of 7.89 kg·cm^−2^ even after 48 h. Following closely behind was Granny Smith, which recorded the second highest level of firmness.

The comprehensive evaluation of firmness and weight loss in the four apple cultivars after storage at different temperatures provides valuable insights into the quality degradation process. The observed decline in firmness across all temperatures, with the highest rate at 8 °C and the lowest at 2 °C, emphasizes the adverse effects of higher storage temperatures on apple quality. The initial firmness measurements, coupled with subsequent assessments over time, effectively demonstrated the degradation of quality across all four cultivars. It is noteworthy that storage temperature exerted a substantial influence on firmness, serving as an indicator of quality deterioration. 

These findings hold significant implications for the apple industry, enabling stakeholders such as growers, distributors, and retailers to make informed decisions regarding postharvest handling practices. By recognizing the critical role of storage and transportation temperature in maintaining firmness and minimizing weight loss, industry can significantly benefit. However, cost of maintaining the lower storage temperature also need to be taken into consideration. 

To further explore the relationship between temperature and the rate of firmness and weight loss, the slopes of the trends depicted in Figure 3 and Figure 4 were calculated for each cultivar across the storage temperatures of 2 °C, 4 °C, 6 °C, and 8 °C. These slopes provide valuable insights into the rate at which firmness and weight are affected by temperature variations. The results of these calculations are plotted in Figure 5 and Figure 6, showcasing the rate of firmness and weight loss with respect to temperature for each cultivar. The coefficient of determination (R^2^) for each relationship is indicated in both figures, providing an assessment of the strength of the correlations.

Figure 5 demonstrates the rate of firmness change as a function of temperature, illustrating the varying slopes for each cultivar. The R^2^ values associated with these relationships further confirm the significance of temperature in influencing firmness degradation. Meanwhile, Figure 6 presents the rate of weight loss with temperature for the different cultivars. As observed, the slopes vary among the cultivars, indicating different rates of weight loss in response to temperature fluctuations. The coefficient of determination (R^2^) values in Figure 6 quantify the extent to which temperature affects weight loss for each cultivar.

Considering that firmness and weight loss are influenced by both storage time and temperature, it becomes imperative to develop an empirical prediction model that encompasses both factors. By utilizing the experimental datasets obtained for firmness and weight loss, a prediction model can be established to estimate the impact of time and temperature on these quality indicators.

The development of such a model holds considerable value, as it would enable industry professionals to make informed decisions regarding postharvest handling practices. By employing the empirical prediction model, stakeholders in the apple industry would gain the ability to optimize storage conditions, anticipate the rate of firmness and weight loss, and enhance overall product quality and shelf life. The experimental data sets serve as the foundation for this predictive model, allowing for accurate estimations and insights into the interplay between storage time, temperature, and the resulting changes in firmness and weight loss.

The general formula for the multiple regression model is:Y = b_0_ + b_1_X_1_ + b_2_X_2_ + ……b_n_X_n_(6)

For firmness function f(t, T):F = a + αt + βT(7)
where F is the firmness, a is constant, α is the estimated regression coefficient of time t, and β is the estimated regression coefficient of temperature T. 

After fitting the data into regression model using SPSS software, version 28.0.1.0 (142), the firmness function of time and temperature is as follows:F(PL) = 8.98 − 0.023t − 0.061T(8)
F(GS)= 8.952 − 0.028t − 0.076T(9)
F(RG) = 8.44 − 0.019t − 0.069T(10)
F(RD) = 8.11 − 0.016t − 0.053T(11)

Similar to the firmness fitting, the weight loss is also predicted using the linear regression model. The function of weight loss (WL) based on both temperature and time are as the following equations:W(GS) = −0.087 + 0.003t + 0.021T(12)
WL(PL) = −0.262 + 0.006t + 0.058T(13)
WL(RG) = −0.274 + 0.006t + 0.056T(14)
WL(RD) = −0.128 + 0.005t + 0.037T(15)

### Validation of the Models

The proposed models were validated using a new set of experimental data using apples stored at a different temperature (10 °C) with the same time durations. The results from the experiments were plotted and compared with the results from the predicted models from Equations (8)–(15) (Figure 7 and Figure 8). 

Figure 7 is dedicated to illustrating the observed changes in firmness for each apple cultivar during a 48-h storage period at a temperature of 10 °C. Similarly, Figure 8 presents the plots showcasing the changes in weight loss for each cultivar under the same storage conditions. The data points represent the actual measurements obtained from the experiment, while the solid lines on the graphs represent the predictions derived from the multi-regression model based on Equations (8)–(15). The legend accompanying the figures provides information about the experimental conditions corresponding to each dataset. The inclusion of the prediction lines allows for a visual comparison between the observed and predicted values, enabling a comprehensive assessment of the model’s accuracy.

In order to validate the accuracy of the multi-regression model, a comparison was made between the predicted values and the experimental values for both firmness and weight loss as the validation process is crucial in assessing the reliability and effectiveness of the model. The results of this comparison were plotted in Figure 9 and Figure 10, respectively, for all the apple cultivars at the temperature of 10 °C.

Upon inspection of the plotted data, a clear and close fit between the predicted and experimental values can be observed in both Figure 9 and Figure 10. The data points align closely along the diagonal lines with a slope of 1.0, indicating a high degree of agreement between the predicted and experimental values. The linear regression equation derived from the firmness values yielded y = 1.0936x − 0.7377, with a slope of 1.0939 that is very close to 1.0. This close proximity to 1.0 indicates a highly accurate prediction of firmness. The coefficient of determination (R^2^) was calculated to be 0.9544, highlighting the model’s ability to accurately predict firmness under the tested conditions.

Similarly, the regression equation for weight loss, derived from the combined data of all apple cultivars, was found to be y = 1.007x − 0.00003. The slope of the equation was also very close to 1.0, reinforcing the model’s accuracy in predicting weight loss values. The coefficient of determination (R^2^) obtained for the weight loss predictions was 0.9964, signifying an exceptional level of accuracy in the model’s predictions.

From the figures it can noticed that the weight loss data shown better correlation than firmness data. Firmness is affected more by the transportation condition, as well as the maturity level at harvest is affecting firmness, and could be the reason behind the fitting of data [19]. Overall, the validation process demonstrates the robustness and reliability of the multi-regression model in predicting both firmness and weight loss for the tested apple cultivars at a temperature of 10 °C. The close fit between the predicted and experimental values, as evidenced by the alignment along the diagonal lines with a slope of 1.0, underscores the model’s high level of accuracy and its ability to provide precise predictions for quality indicators under the specified storage conditions.

## 4. Conclusions

The objective of this study is to examine how storage conditions and duration affect some physical quality attributes of apple cultivars, and to develop empirical prediction models for weight loss and firmness under controlled temperature during transportation and short-term storage. The findings suggest that the firmness and weight loss is significantly impacted by storage temperature. Lowering the temperature can minimize changes in firmness and weight loss, thereby preserving the quality of the product during transportation and storage. 

The empirical models developed in this study demonstrate a robust relationship between predicted and experimental results at 10 °C. These models can be utilized by producers and retailers to forecast product physical quality under different circumstances, enabling them to adjust storage and transportation conditions to maintain product quality over an extended period. Future research should focus on evaluating non-invasive microimaging techniques for assessing apple firmness to achieve more accurate outcomes. Furthermore, the models could be enhanced by incorporating relative humidity levels and their impact on the quality parameters examined. Although higher quality can be achieved by lowering the storage temperature, it also increases the storage cost. Future studies can be conducted to determine the optimum storage conditions taking quality target and cost into consideration. More studies are also required to investigate other quality attributes, including biochemical properties. 

## Figures and Tables

**Figure 1 foods-12-02113-f001:**
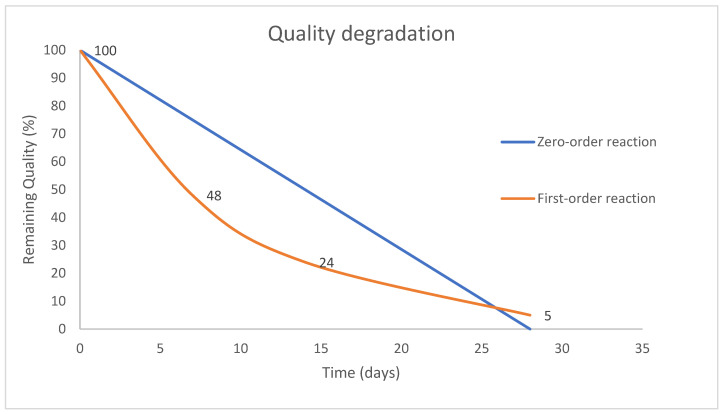
Quality degradation model.

**Figure 2 foods-12-02113-f002:**
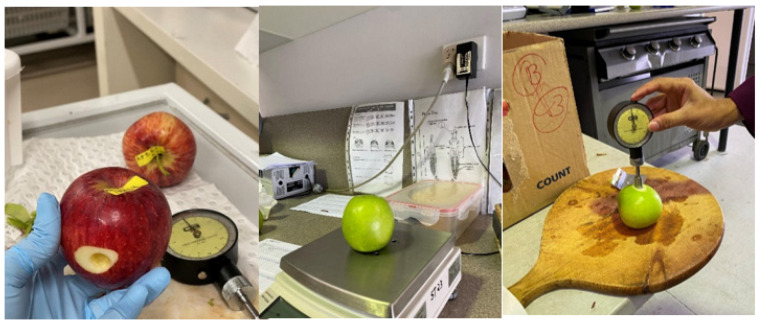
Experimental procedure.

**Figure 3 foods-12-02113-f003:**
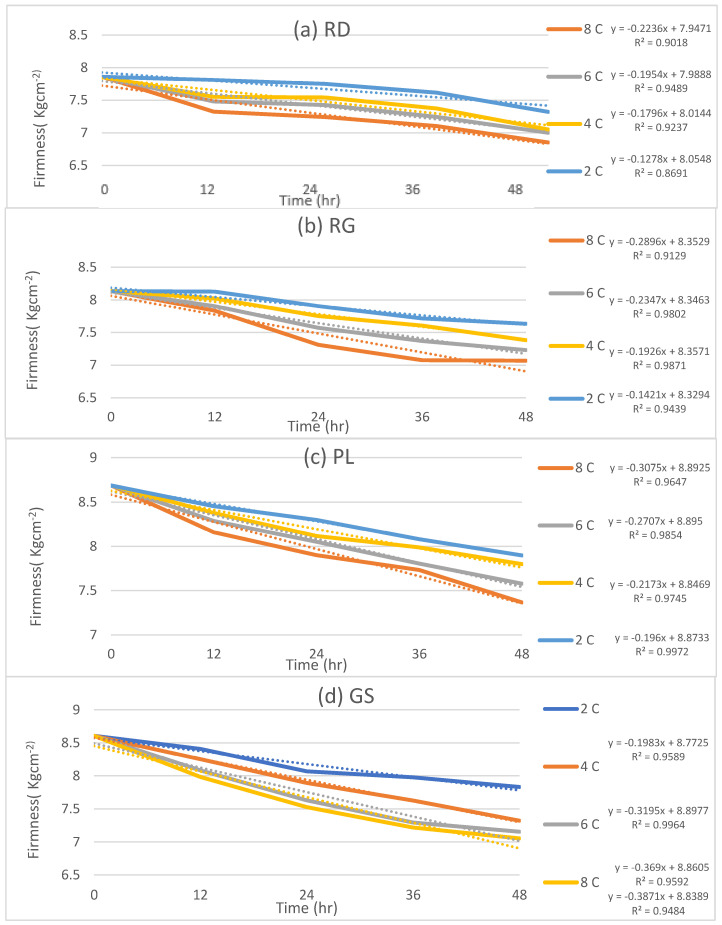
Change in firmness with time at different temperatures (**a**): Red Delicious, (**b**): Royal Gala, (**c**): Pink Lady, (**d**): Granny smith.

**Figure 4 foods-12-02113-f004:**
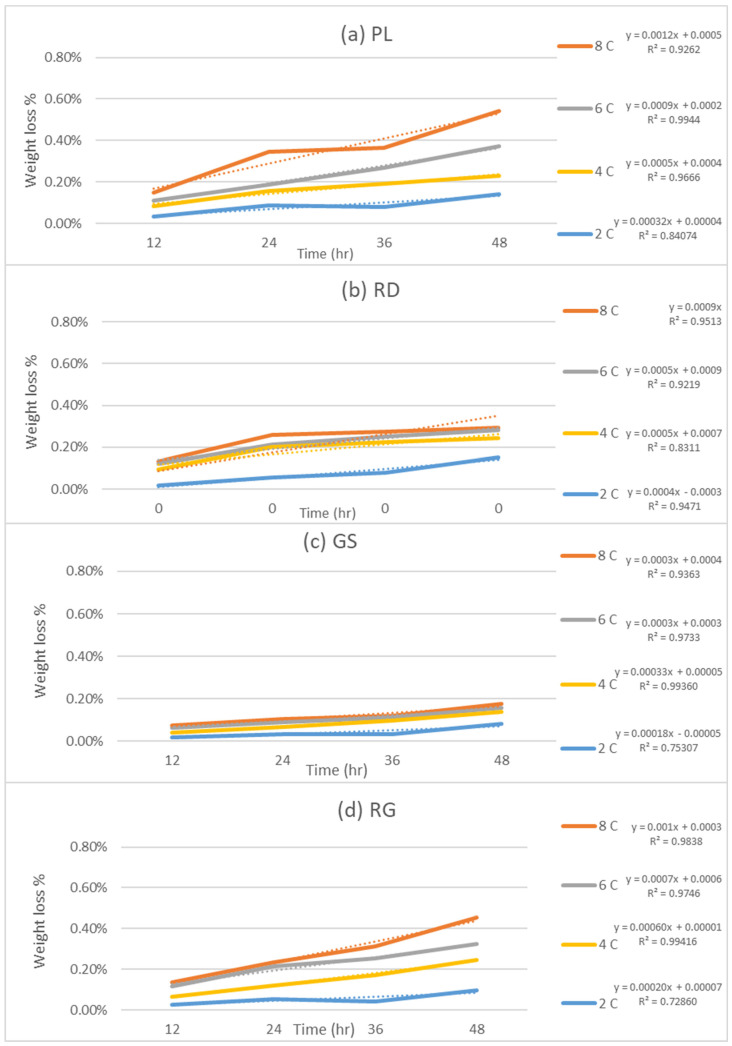
Change in weight loss with time at different temperatures (**a**): Pink Lady, (**b**): Red Delicious, (**c**): Granny Smith, (**d**): Royal Gala.

**Figure 5 foods-12-02113-f005:**
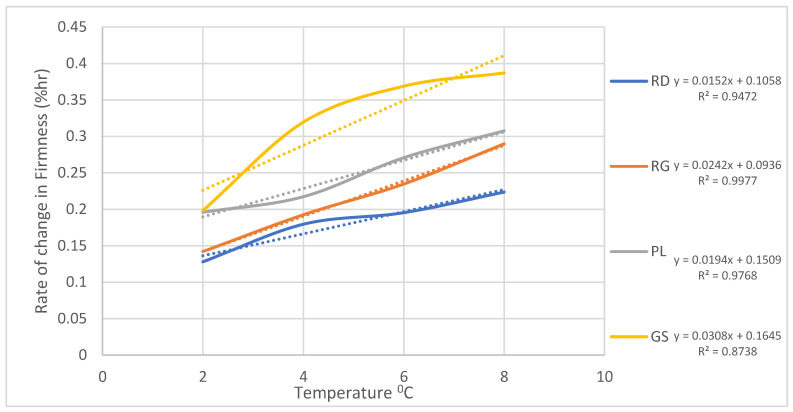
Rate of firmness loss as a function of temperature.

**Figure 6 foods-12-02113-f006:**
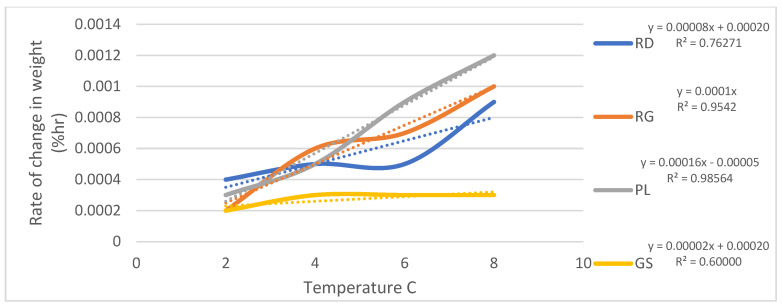
Rate of weight loss as a function of temperature.

**Figure 7 foods-12-02113-f007:**
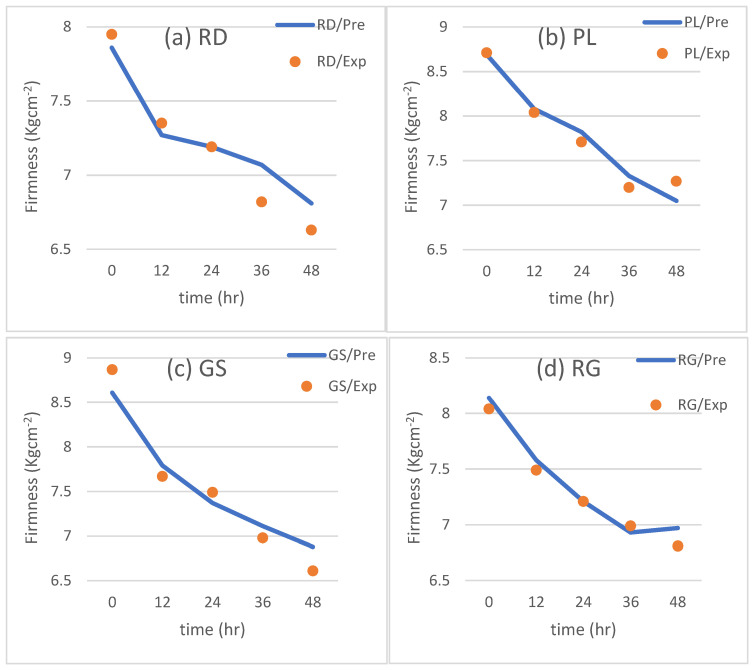
Validation of the firmness model: (**a**) RD, (**b**) PL, (**c**) GS, (**d**) RG.

**Figure 8 foods-12-02113-f008:**
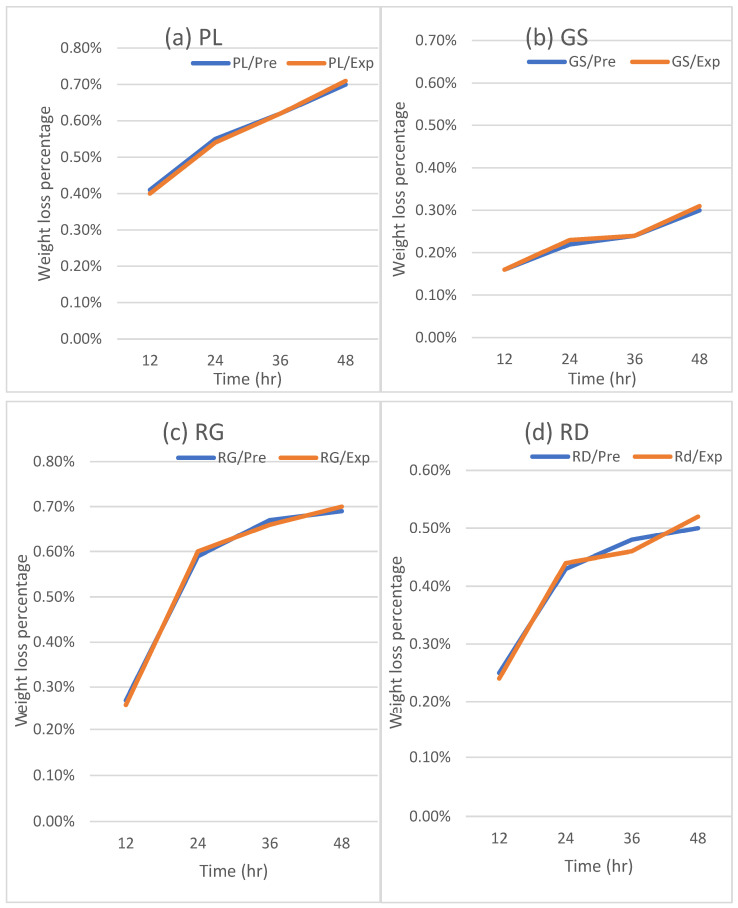
Validation of the weight loss model: (**a**) PL, (**b**) GS, (**c**) RG, (**d**) RD.

**Figure 9 foods-12-02113-f009:**
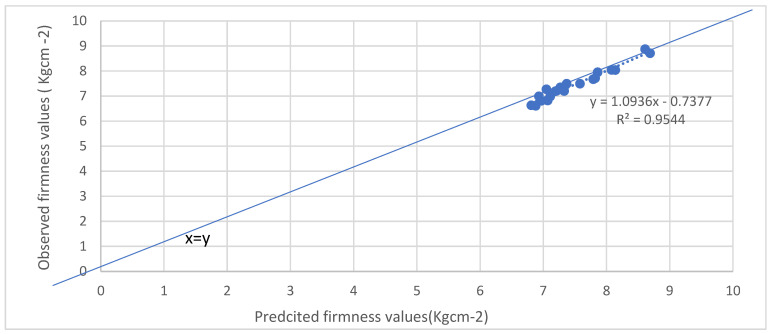
Predicted versus observed firmness values for all cultivars.

**Figure 10 foods-12-02113-f010:**
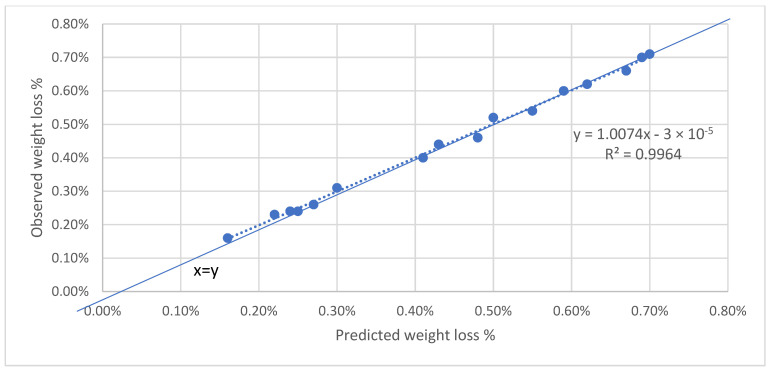
Predicted versus observed weight loss values for all cultivars.

## Data Availability

Data is contained within the article.

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
