# Peer review of "An Empirical Model for Predicting the Fresh Food Quality Changes during Storage"

_foods, 2023, doi:10.3390/foods12112113_

Round 1

Reviewer 1 Report

The manuscript Experimental Investigation and Empirical Model Development for Quality Changes during Storage  conducted on the changes in quality attributes of four fresh apple cultivars (Granny Smith, Royal Gala, Pink Lady, and Red Delicious) during transportation and storage. The study evaluates the weight loss and firmness of these apples at different cooling temperatures ranging from 2 C to 8 C to assess the impact of storing at these temperatures on the quality attributes. 

The manuscript is prepared professionally. It includes a well-crafted abstract and an exhaustive introduction that justifies the research undertaken. The introduction points to the deficiencies in the literature on the subject. The aim is clearly defined. Modern analytical methods were used in the research. The discussion of the results is well prepared. The conclusions are well-defined. The illustrative material is appropriate.

In my opinion, the manuscript after corrections, will be suitable for publication in a journal.

Detailed comments:

Abstract: Too short and must be increased (enlarged) by using obtained numeric data from results.

Do not use abbreviations when use first time.

Introduction - The introduction is enough in my opinion. Introduction needs some modifications.

The global supply chains for fresh food have been evolving significantly in response to a growing emphasis on maintaining high quality standards. The impetus for this transformation has largely been driven by competitive strategic decisions and the stringent quality requirements of supermarkets. Producers and distributors are currently prioritizing the preservation of fruit quality during the transportation and storage phases of the supply chain. This is being achieved through the utilization of advanced cooling and preservation technologies aimed at ensuring the delivery of fresh and high-quality fruits to the market (Ali et al., 2021; Abanoz and Okcu, 2022; Saleem et al. 2022; Saran et al., 2022)

I suggest below references

Ali, S.; Anjum, M.A.; Ejaz, S.; Hussain, S.; Ercisli, S.; Saleem, M.S.; Sardar, H. Carboxymethyl cellulose coating delays chilling injury development and maintains eating quality of ‘Kinnow’ mandarin fruits during low temperature storage. Int. J. Biol. Macromol. 2021, 168, 77–85.

Abanoz, Y.Y.; Okcu, Z. Biochemical content of cherry laurel (Prunus laurocerasus L.) fruits with edible coatings based on caseinat, Semperfresh and lecithin. Turk. J.  Agric. For. 2022, 46 (6), 908-918.

Saleem, M.S.; Ejaz, S.; Anjum, M.A.; Ali, S.; Hussain, S.; Ercisli, S.; Ilhan, G.; Marc, R.A.; Skrovankova, S.; Mlcek, J. Improvement of Postharvest Quality and Bioactive Compounds Content of Persimmon Fruits after Hydrocolloid-Based Edible Coating Application. Horticulturae 2022, 8, 1045.

Saran, E.Y.; Cavusoglu, S.; Alpaslan, D.;  Eren, E.;Yilmaz, N.; Uzun, Y. Effect of egg white protein and agar-agar on quality of button mushrooms(Agaricus bisporus) during cold storage. Turk. J. Agric. For. 2022, 46 (2): 173-181.

Please divide below sentence to 2-3 small sentences and use new references

The global supply chain for fresh fruits and vegetables is undergoing a significant transformation, moving away from the traditional model that involved intermediaries such as wholesalers and towards a more direct and collaborative relationship between suppliers and supermarkets. This shift allows supermarkets to establish a reliable supply of fresh produce directly from farms, ensuring consistent quality. To strengthen this direct relationship and build consumer trust, supermarkets have introduced private regulatory standards that provide greater quality assurance for their fresh produce supply (Spencer, 2004). In Europe, these standards are often developed through stakeholder coalitions to ensure uniform criteria across supermarkets. In Australia, the two main supermarket chains, Coles and Woolworths, have each established their own private regulatory standards – Coles Supplier Requirements (CSR) and Woolworths Supplier Excellence Program (WSEP) – to regulate the quality of fresh produce (Davey & Richards, 2013).

The manuscript Experimental Investigation and Empirical Model Development for Quality Changes during Storage  conducted on the changes in quality attributes of four fresh apple cultivars (Granny Smith, Royal Gala, Pink Lady, and Red Delicious) during transportation and storage. The study evaluates the weight loss and firmness of these apples at different cooling temperatures ranging from 2 C to 8 C to assess the impact of storing at these temperatures on the quality attributes. 

The manuscript is prepared professionally. It includes a well-crafted abstract and an exhaustive introduction that justifies the research undertaken. The introduction points to the deficiencies in the literature on the subject. The aim is clearly defined. Modern analytical methods were used in the research. The discussion of the results is well prepared. The conclusions are well-defined. The illustrative material is appropriate.

In my opinion, the manuscript after corrections, will be suitable for publication in a journal.

Detailed comments:

Abstract: Too short and must be increased (enlarged) by using obtained numeric data from results.

Do not use abbreviations when use first time.

Introduction - The introduction is enough in my opinion. Introduction needs some modifications.

The global supply chains for fresh food have been evolving significantly in response to a growing emphasis on maintaining high quality standards. The impetus for this transformation has largely been driven by competitive strategic decisions and the stringent quality requirements of supermarkets. Producers and distributors are currently prioritizing the preservation of fruit quality during the transportation and storage phases of the supply chain. This is being achieved through the utilization of advanced cooling and preservation technologies aimed at ensuring the delivery of fresh and high-quality fruits to the market (Ali et al., 2021; Abanoz and Okcu, 2022; Saleem et al. 2022; Saran et al., 2022)

I suggest below references

Ali, S.; Anjum, M.A.; Ejaz, S.; Hussain, S.; Ercisli, S.; Saleem, M.S.; Sardar, H. Carboxymethyl cellulose coating delays chilling injury development and maintains eating quality of ‘Kinnow’ mandarin fruits during low temperature storage. Int. J. Biol. Macromol. 2021, 168, 77–85.

Abanoz, Y.Y.; Okcu, Z. Biochemical content of cherry laurel (Prunus laurocerasus L.) fruits with edible coatings based on caseinat, Semperfresh and lecithin. Turk. J.  Agric. For. 2022, 46 (6), 908-918.

Saleem, M.S.; Ejaz, S.; Anjum, M.A.; Ali, S.; Hussain, S.; Ercisli, S.; Ilhan, G.; Marc, R.A.; Skrovankova, S.; Mlcek, J. Improvement of Postharvest Quality and Bioactive Compounds Content of Persimmon Fruits after Hydrocolloid-Based Edible Coating Application. Horticulturae 2022, 8, 1045.

Saran, E.Y.; Cavusoglu, S.; Alpaslan, D.;  Eren, E.;Yilmaz, N.; Uzun, Y. Effect of egg white protein and agar-agar on quality of button mushrooms(Agaricus bisporus) during cold storage. Turk. J. Agric. For. 2022, 46 (2): 173-181.

Please divide below sentence to 2-3 small sentences and use new references

The global supply chain for fresh fruits and vegetables is undergoing a significant transformation, moving away from the traditional model that involved intermediaries such as wholesalers and towards a more direct and collaborative relationship between suppliers and supermarkets. This shift allows supermarkets to establish a reliable supply of fresh produce directly from farms, ensuring consistent quality. To strengthen this direct relationship and build consumer trust, supermarkets have introduced private regulatory standards that provide greater quality assurance for their fresh produce supply (Spencer, 2004). In Europe, these standards are often developed through stakeholder coalitions to ensure uniform criteria across supermarkets. In Australia, the two main supermarket chains, Coles and Woolworths, have each established their own private regulatory standards – Coles Supplier Requirements (CSR) and Woolworths Supplier Excellence Program (WSEP) – to regulate the quality of fresh produce (Davey & Richards, 2013).

Reviewer 2 Report

There are some comments which can help improve the paper: -Title is not appropriate. -In the introduction section is very long and many of the sentences presented lack references, it is best to use new and related citations.

-In the materials and methods section, the attributes evaluated do not truly contribute to the conclusion of the manuscript, since the number of attributes evaluated for a full research paper is quite limited. It would be better to also evaluate the biochemical properties.

-At the end of introduction, the aim of the study should be more detailed, and novelty is questionable. Justify novelty in Introduction section.

-In the figure, the significant and non-significant letters or the SE should be added.

-In the manuscript text C, ⁰C,  ⁰ degrees Celsius  or Celsius hip? Which is correct? Please check the whole manuscript and correct them.

- Figure 3, (a) axis x or during storages (time) should be corrected.

-Discussion is not 'in depth’ and incorporates some recent literature, the discussion needs to be rewritten in context of the prescribed aims of the manuscript.

-Style of writing of the references is not uniform.

-Major revisions of English are suggested.

-My recommendation is "Reject".

General revisions of English are suggested

Reviewer 3 Report

This paper presents "Experimental Investigation and Empirical Model Development for Quality Changes during Storage".

The manuscript is not carefully drafted. There are many notations in the manuscript for degrees Celsius. Please make it uniform throughout the text. Please use the superscript/subscript in the figures (e.g. 3, 7, 9) for the units of measure on the 0Y axis. There are few bibliographical references. More recent literature must be studied and new bibliography introduced.

Round 2

Reviewer 2 Report

 -The introduction section is long and needs to be shortened, and the section does not need to be divided into sections.  It is also necessary to include some recent references (2022 and 2023).

 -The figures are not of good quality.

-Why are some data missing SD values in Table 2A?

Weight should be written in lowercase letters in Table 2 title.

-Minor revisions of English are suggested.

-Minor revisions of English are suggested.
